# Nutritional Habits of Hungarian Older Adults

**DOI:** 10.3390/nu16081203

**Published:** 2024-04-18

**Authors:** Rita Soós, Csilla Bakó, Ádám Gyebrovszki, Mónika Gordos, Dávid Csala, Zoltán Ádám, Márta Wilhelm

**Affiliations:** 1Doctoral School of Health Sciences, Faculty of Health Sciences, University of Pécs, 7621 Pécs, Hungary; soosrita8@gmail.com; 2Institute of Sport Sciences and Physical Education, Faculty of Sciences, University of Pécs, 7624 Pécs, Hungary; bakocsillairoda@gmail.com; 3Doctoral School of Biology and Sportbiology, Faculty of Sciences, University of Pécs, 7624 Pécs, Hungary; gyebrovszkiadam@gmail.com (Á.G.); fm.csaladavid@gmail.com (D.C.); 4Department of Pharmaceutical Biotechnology, Faculty of Pharmacy, University of Pécs, 7624 Pécs, Hungary; adam.zoltan.mihaly@pte.hu; 5Wnt-Signaling Research Group, Szentagothai Research Center, University of Pécs, 7624 Pécs, Hungary

**Keywords:** nutritional habits, elderly, macronutrient consumption, malnutrition, fruit and vegetable consumption

## Abstract

There are many nutritional changes that come with aging, mostly as consequences of health regression. Malnutrition and overweight often start with inadequate food consumption, followed by alterations in biochemical indices and body composition. In our study, we aimed to analyze the feeding habits and energy and nutrient intake of a Hungarian elderly population, focusing on macronutrient, water, fruit, and vegetable consumption while searching for possible nutritional factors leading to NCD and many other chronic diseases in this population. Two questionnaires were used. These were the Mini Nutritional Assessment (MNA) and one asking about nutritional habits, and a 3-day feeding diary was also filled. Subjects (*n* = 179, 111; females (F), 68 males (M), older than 50 years were recruited. Based on MNA results, 78 adults (43.57% of the studied population) were malnourished or at risk of malnutrition, although, according to BMI categories, 69% were overweight and 7.3% were obese among M, while 42.3% were overweight among F. The average daily meal number was diverse. The amount of people consuming fruit (11.7%) and vegetables (8.93%) several times a day was extremely low (15.3% of F and 4.4% of M). Daily fruit consumption in the whole sample was 79.3%. Overall, 36.3% consumed 1 L of liquid and 0.5 L of consumption was found in 15.1% of participants. A significant gender difference was found in water consumption, with F drinking more than M (*p* ≤ 0.01). In our sample, 27.93% of the respondents took dietary supplements. Further analysis and research are needed to explore the specific health implications of and reasons behind these findings.

## 1. Introduction

Noncommunicable diseases (NCD) are leading causes of death worldwide [1]. The most important reasons behind them are physical inactivity and poor diet. The nutritional habits of subjects nowadays are strongly related to free sugar consumption. Higher sugar intake will increase calorie intake, the probability of overweight, obesity, and the development of noncommunicable diseases [1], just as socioeconomic status is one of the strongest predictors of obesity and unhealthy food environments [2]. Free sugar consumption and NCD are higher in the low- and middle-income countries (LMICs), like Hungary. According to dietary guidelines, added sugars should be ˂10% of calories consumed (US Dep), and an association between consumed sugars and the quality of nutrition has been shown, especially in the case of micronutrients [3]. Interestingly, younger subjects consume more added sugar than the 51+ age group in the USA, Australia, and Sweden [3]. The increase in free sugar consumption correlated with the reduction in several micronutrients, especially in the younger population, while in older adults, vitamin D consumption was significantly decreased with levels of higher sugar consumption. There are many studies discussing socioeconomic determinants of dietary patterns [4]. In LMICs, there was a strong correlation between fruit and vegetable intake, just as with healthy food consumption and the income of the population.

There are many changes in physiologic functions, nutritional status, and diet that come with aging. It is not clear whether changes in physiological functions, nutrition, are inherent to the aging process or whether they reflect changes in physical activities, diet, body composition, or chronic illness [5]. The development of malnutrition often starts with inadequate food consumption, followed by signs of alterations in biochemical indices and body composition. There is, however, no “gold standard” for the assessment of malnutrition. However, nutritional assessments help in the evaluation of feeding problems that lead to clinical complications [6]. In a Turkish and Portuguese nutritional study of elderly populations, authors describe differences in terms of malnutrition related to sociodemographic factors [7]. Among these, income and educational status are important, as well as whether people live alone or in families. Osteopenia and osteoporosis are also important consequences of aging, and can be related to unfavorable lifestyle habits. In a Chinese postmenopausal population, lean mass was the best determinant of bone mineral density [8]. Six dietary patterns were also identified, but only cereal grains–fruit and milk–root vegetable patterns were associated with bone the density of the spine and hip.

Obesity is recognized as a major risk factor in the development of cardiovascular problems and diabetes, but in several chronic diseases a higher BMI may be associated with a lower mortality and better outcomes compared with their normal-weight counterparts. This “obesity paradox” was mostly reported in the elderly. Lainscak et al. [9] evaluated nine large-scale studies of the obesity paradox in chronic diseases. The paradox (especially above the age of 62 years) might be partly explained by the fact that BMI does not differentiate between lean body mass and fat mass [10], but is only an indicator of the feeding status of a subject. Higher mortality in the low-BMI categories may be due to sarcopenic obesity, which is characterized by low muscle mass and strength, while fat mass may be preserved or even increased. Sarcopenia exacerbates insulin resistance and dysglycemia in both nonobese and obese individuals, therefore increasing the risk of adverse effects, such as physical disability, poor quality of life, and death. So, in the case of the elderly, it is better to evaluate functional status based on the amount of muscle mass and its function (mainly strength and physical performance) than based on BMI. Using BMI as the only indicator of nutrition would fail to identify nutritional issues in these individuals [11].

The maintenance of a proper water balance is fundamental for maintaining health or a healthy lifestyle. The water needs of a sedentary adult are about 1.5 L daily, and they are essential for hydration [12] and maintaining normal physiological functions. This role of hydration is increasingly recognized. Intake requirements vary across the whole population depending on physical activity, environmental conditions, dietary patterns, alcohol intake, health problems, polypharmacy, and age. Elderly individuals have a higher risk of dehydration. Lower water consumption and increased liquid losses both cause dehydration in this population [13]. Dehydration affects 20–30% of older adults, leading to increased mortality, morbidity, and disability even more than in younger adults [14].

Fruits and vegetables are very important in each age group. The increased elderly consumption of fruits and vegetables significantly reduces the risk of cognitive decline [15,16]. In the case of high fruit and vegetable consumption, combined with high physical activity, the risk of cognitive decline is reduced by more than half (63%) [16]. Many meta-analyses have also indicated that fruit and vegetable consumption decreases the risk of diseases such as cancer, stroke, diabetes, hypertension, and heart diseases [17]. Most health guidelines recommend a minimum of five servings of fruit and vegetables per day, although these suggestions vary a lot. The Eurodiet core report and the World Cancer Research Fund recommends an intake of at least 400 g/day, while the suggestion in Denmark is 600 g/day, and in the USA it is 640–800 g/day [17]. For the elderly, the consumption of quality nutrients may be even more important in terms of preventing chronic diseases, resulting from malnutrition, or a diet that is typically missing micro- and macronutrients [18]. Since free sugar consumption strongly correlates to micronutrient amounts in the elderly, fruit and vegetable consumption amounts are usually lower in this population, and the usage of dietary supplements plays an important role. In sarcopenia, the general recommendation is that patients consume whey protein combined with resistance training to increase protein synthesis, thus avoiding muscle loss [19]. In order to slow down the deterioration of cognitive abilities, it is recommended to consume vitamins B6 and B12, as these reduce serum homocysteine levels, consequently reducing the risk of Alzheimer’s disease [20]. 

In the Hungarian population, neither nutrition nor physical activity status seem to meet the European standards. Füzesi et al. (2004) [21], in a large-scale study, described with the aid of SF36 how the age group between 55 and 64 years has very low scores in terms of overall health, physical functions, physical role, pain, and vitality. The measured population reported very low levels of daily physical activities (35% or more doing nothing, not even walking). Considering BMI categories, most of the overweight and obese population were older, having the lowest general health and physical functions. Subjects in the underweight category were much younger (30.82 y average), having the same scores in general health, vitality, and physical function as seen in the normal BMI range. In a survey measuring the dietary habits and activity of elderly Hungarians [22], nearly 60% of people had three meals per day, a rate which increased with aging. The most substantial meal was lunch; meals were prepared at home by 90% of the elderly. Lard was used for cooking by 44% of subjects. Milk, dairy products, fish, fruits, fresh vegetables, and vegetable dishes were consumed at a rate far below the recommendations. All types of meat consumption were radically decreased compared to former official Hungarian surveys before, especially beef. Legumes were consumed several times within a week. The average time spent performing outdoor activities was 12 h per week. These consisting of walking, gardening, shopping, but not exercise. According to official statistical data, the quality-of-life index scale was found to be quite low in Hungary (2023–2024 data) compared to other European countries, and a low purchasing power (49.8) and a high ratio of property price to income were reported [23]. Within Hungary, the region south of the Danube had lower incomes. Unfavorable changes were seen compared to previous a nationwide survey (from 2009 to 2014), with an increase in fat and saturated fatty acid energy percentage in women, and a decrease in fruit and vegetable consumption, explaining the decreased fiber intake. An increasing trend was observed compared to 2009 in terms of added sugar energy percentage in each age groups of both genders; meanwhile, the frequency of having meals also decreased [24]. A similar trend was found in fat consumption in another national survey, which mentioned an increased fat amount and higher cholesterol levels [25].

In our study we aimed to measure (i) the feeding habits of the Hungarian elderly population, focusing on water, fruit, and vegetable consumption, in order to analyze the possible nutritional factors leading to NCD and many other chronic diseases in this population. We aimed (ii) to assess and analyze the nutritional status, malnutrition risk, and food consumption frequency of people over the age of 50 years living in their own homes. The examination of the prevalence of obesity and malnutrition was based on BMI and MNA (Mini Nutritional Assessment) questionnaires performed in the studied population, looking for gender differences. Energy and nutrient consumption were analyzed after self-reports around diet, focusing on macronutrient (protein, carbohydrate, and fat), water, fruit, and vegetable consumption.

## 2. Subjects and Methods

In our study, subjects living in their own homes and older than 50 years were recruited and divided into 8 age groups based on years. The questionnaires were spread in a directed way (in printed form) among elderly people regularly attending routine medical checkups, with the hope that at least 6 persons would respond in each age group. The subjects included in this program were younger than would be expected, since it was demonstrated from a representative sample that biological aging and regression is faster in the Hungarian population, especially among females [21]. 

Exclusion criteria were dementia, bad health status/sickness with special dietary needs, dialysis, or special feeding through the nose. Those who had major sensory disabilities (poor hearing, or vision) were also excluded from this study. 

Data collection was conducted with the intention of collecting higher numbers of responders with useful and trustful results, so helpers were trained (nurses and physiotherapists, *n* = 10) to ask and code the answers in the same way using the snowball method. Trained helpers also explained personally all the important questions and facts of the questionnaires, describing the possible answers in detail, helping respondents to fill them in correctly and trustfully.

Two questionnaires designed for the elderly were used: The standardized Mini Nutritional Assessment (MNA) [26], helping to measure the feeding status of elderly subjects. In a clinical setting, it was found to be very sensitive and specific (96%) and also reliable (98%) [27]. The MNA is the result of a geriatric and nutritional project, having 18 simple and brief questions (6 lifestyle, motor abilities, and medicational status; 8 for nutritional habits, including feeding, water consumption, and self-management in terms of nutrition; and 4 subjective questions on self-perception in health and nutrition). With its aid, it is possible to identify people at risk of malnutrition—indicated by scores between 17 and 23.5—before severe changes in weight or albumin levels occur. These individuals are likely to have a decreased caloric intake that can be corrected by nutritional interventions.

The second questionnaire was a Food Frequency Questionnaire, including 4 questions about biodata, and 13 about demographic and nutritional habits (like the frequency of eating, drinking, who makes the meal etc.). Next, a table was filled out, asking the frequency of 54 types of food consumption (types of answers: never, rarely, once a month, once a week, daily, and several times a day). Types of food included brown or white bread, mueslis, noodles, pastries, rice, potato, milk, yoghurt, kefir, cheese, eggs, fruits (fresh or frozen), vegetable, pickles, legumes, nuts, meat types, fish, water, tea, or coffee, juices, snacks, etc. Finally, based on the frequency of protein and CH consumption (categorized by the filled food table), cumulative points were also created. In CH consumption, the possible maximal points were 25, while the calculation of protein consumption saw a maximum of 30 points added. The questions asked were matched to Hungarian food types, helping respondents to identify them, and we kept in mind that possible wrong answers could cause misinterpretation. As such, the frequency questionnaire matched the standards mentioned by [28]. 

The third part of this survey contained a 3-day diet diary. We tracked two non-consecutive weekdays and one weekend day was requested [29]. All the subjects underwent training before the start of data collection, teaching the strategy of categorization. Subjects had to note each meal or drink, snack, etc., separately in each day’s preformed sheet, describing the type and amount of food or drink. The weight and the type of the food (if prepacked, or sliced) were also asked, just like the number of cooked meals (1, or 2 plates). In the case of liquid consumption, it was expected of the participants to give their answers in mL, or cups. Completing the survey took 15–20 min, and after 3 days the diary was collected. All the questionnaires and diaries were collected for data analysis. The consumed carbohydrate, protein, fat amount, and calories were calculated for each day. Since traditional Hungarian dishes were also part of foods consumed, calories were calculated with the aid of www.caloriabazis.hu [30] calculator. This software uses the USDA (National Nutrient database for Standard Reference). Analyzing the frequency of food consumption survey, two major datasets were created, namely protein and carbohydrate groups. The main question was how to determine the ratio of these products in the diet of Hungarian elderly. Answers (never, rarely, etc.) were converted into points, the maximal point for carbohydrate consumption was 25, and in the case of protein it was 30. The average carbohydrate-to-protein ratio was calculated for each age group.

Finally, 179 responses were analyzed (females (F): 111, males (M): 68). The ratio of genders was different, since males (similarly to other Hungarian studies) were not cooperative in this study either. The age distribution of responders was as follows: 50–55 years: 22 F, 10 M; in each group of 56–60, 61–65, 66–70 and 71–75 years, 15 F and 10 M responded. In the 76–80 years age group, 15 F and only 6 M cooperated; in the population of 81–85, 8 F and 6 M replied; and above 85 years, 6 F and 6 M were cooperative. It is important to note that life expectancy at birth in Hungary is below 75 years, and it is difficult to find responders above this age group. Some 44% of subjects lived in cities, 51% in county seats, and only 5% lived in villages. None of the responders were vegans or reported following vegetarian diets.

## 3. Statistical Analyses

Besides descriptive statistics, linear regression and a Khi2 probe were performed. The confidence interval was 95%, and significant differences were recognized if *p* ˂ 0.05.

## 4. Results

A Mini Nutritional Assessment (MNA) is a validated nutrition screening and assessment tool used for identifying geriatric patients 65 years and above who are malnourished or at risk of malnutrition. The average MNA score was 25.28 ± 5.07 in the M population and 22.87 ± 4.74 in the F group.

Based on MNA results, 78 adults (43.57% of the studied population) were malnourished or at risk of malnutrition. There was a significant difference in the risk of malnutrition between M and F (Table 1). Females had a risk of malnutrition 4 times higher than males (OR = 3.96; CI95% = 1.83–8.55). Gender differences were found in the number of subjects having total MNA score of ≤17, and females were more numerous in this case (F = 18, M = 5). The loss of appetite and weight loss was examined with the aid of the questionnaire and 3 groups were created. Overall, 23.5% of M responders and 31.5% of F responders reported weight loss, while 73.5% of M and 58.6% of F responders did not lose weight. Other participants did not remember or know. Having no appetite was reported by 22.1% of M and 37.8% of F participants. Females reported appetite loss significantly more than M (*p* = 0.028; χ^2^(1) = 4.837), the possibility of that being twice as high in F (OR = 2.15).

Among F, there was a positive correlation (week, r^2^ = 0.272) between appetite loss and weight decrease. Lost appetite significantly modified weight loss in 27% (˂0.01). Among M, the same relationship was found. However, there was a stronger correlation (r^2^ = 0.702), and the modifying effect of appetite was 70%.

Females were significantly more uncertain in their own judgements concerning their feeding status than M (*p* = 0.006) (Table 2) and the possibility of misjudging was 2.7 times higher (OR = 2.73; CI95% = 1.31–5.7) than in the M sample.

The BMI categories of subjects were diverse. Performing measurements within the same population, the following data were collected (Table 3). The ratio of having BMI > 25 was much higher among M in each age group, and significant difference was found in the 50–60 years population (Table 4). Interestingly enough, in the 61–65 years age, group 100% of M and 73% of F were in the overweight category. Surprisingly, among females, 59% were in the overweight and obese category after BMI calculation. However, analyzing the same population according to the MNA scores, 26% (17 females) of them fell into the category at risk of malnutrition. Among males, 76.3% were overweight or obese, while 9.6% (5 subjects) of them fell into the risk of malnutrition based on MNA scores. In the whole sample, 117 subjects (65%) were overweight or obese and 18.8% (22) of them were at risk of malnutrition.

The average number of daily meals was very diverse. Eating once a day was only found among F (2.7%). Overall, 7.26% of responders reported eating twice a day (F 9.0%; M 4.41%); three meals were consumed by 71.5% of respondents (F 63.06%; M 85.29%); four meals were consumed by 12.29% of the cohort (F 16.21%; M 5.88%); and five meals were eaten by 8.1% of F and 2.94% of M. Six meals were reported in one day in both genders by one person only. The origin of the consumed meal was also asked. Among F, 73.87% prepared/cooked their own food and 6.3% ordered, while among M, 13.23% prepared their own meal and 75% of them reported eating their wife’s cooking. In total, 72% of F ate at home, while among M it was 54%.

The levels of consumed calories, protein, carbohydrate (CH), and fat is summarized in Table 5 according to the 3-day food diary. The average caloric intake of F was 1667 Kcal, protein consumption was 63 g, CH was 205 g, while fat was 60 g. Males took in 2091 Kcal on average within a day, and 78 g of protein, 245 g of CH, and 85 g of fat consumption was calculated.

The average of points collected from average CH consumption in the whole population was 14.84 (out of maximum 25), standing at 15.82 points for M and 13.86 points for F. The average protein consumption was 12.14 (out of 30 points), standing at 13.11 for M and 11.18 points for F. Calculating the ratio of CH, protein, and fat in a regular diet, among F the protein intake within a day was 19.20%, with CH at 62.5% and fat at 18.29%. Males consumed 19.11% of protein on average in one day; CH was at 60.04% and fat was at 20.83%. The ratio of carbohydrate to protein was 3.25 in the F population, and 3.14 among M. Overall, 64.24% of the population ate white bread daily, and 2.79% chose brown bread. In total, 8.37% consumed white bread several times a day. In the F population, 8.1% of responders never consume white bread, while 56.75% eat it daily. Brown bread’s daily consumption was found to be at 17.1% of F, with 1.8% eating it several times a day. Among M, 1 person reported never eating white bread, while 76.47% reported daily consumption. Overall, 14.7% of M reported the daily consumption of brown bread.

Daily fruit consumption in the whole sample was 79.3% (73% among F and 89.7% among M). Overall, 15.3% of F and 4.4% of M consumed fruit several times a day (Figure 1). In total, 66.4% of respondents over 60 years old consumed fruits once a day, and only 11.5% consumed them several times within a day. All told, 83% of subjects around 85 years of age ate fruit once a day, with 8.3% doing so several times a day.

All told, 63.7% of the respondents consumed vegetables once a day, while 9% consumed them several times per day. Some 60.7% of the elderly over 60 years old consumed vegetables once, but only 5.7% consumed them several times a day. Some 66.7% of respondents aged 85 years and up ate vegetables once a day, with 4% eating them several times. Significant differences (*p* < 0.01) were found between the consumption habits of M and F. 

Gender was influential in fruit and vegetable consumption in the elderly (Table 6). The chance of daily fruit intake by F was almost 4 times more (OR = 3.9 CI95% = 1.103–13.917) than by M participants, and 18.5 times greater (OR = 18.5 CI95% = 2.438–140.11) in terms of weekly consumption. 

In terms of vegetables, F had a twice greater chance (OR = 1.94 CI95% = 0.6–6.277) of consuming them daily and were 24 times more likely (OR = 24.815 CI95% = 3.297–186.778) to repeat this consumption several times a week (Table 6).

Dietary supplements (e.g., vitamins, herbal preparations, etc.) were used by 37.8% of women and 11.7% of men (Table 7). In the entire sample, 50 subjects (27.93%) took them and 129 persons (72.06%) did not take dietary supplements.

In the whole sample, daily water consumption was 2 L (36.9%, with F at 33.3% and M at 42.5%). Water intake above 2 L was found in 11.7% of all participants (F at 19%, while no M were found drinking water above 2 L (Table 8). Among the elderly asked, 36.3% consumed 1 L of liquid (F 42.3%, M 26.5%). The consumption of 0.5 L of liquid was found in 15.1% of participants (F 5.4%, M 31%). Significant differences were found between genders in terms of water consumption, with F drinking more than M *p* ≤ 0.01).

According to our data, water intake of 0.5 L was significantly higher among M than that of F (*p* < 0.001; χ^2^(1) = 21.368); however, F subjects marked 1 L of water consumption significantly more often (Table 8). Females had a twice greater chance (OR = 2.040 CI95% = 1.057–3.936) of 1 L water consumption than M.

## 5. Discussion

In our survey, we aimed to measure the feeding habits of the Hungarian elderly population, focusing on water, fruit, and vegetable consumption to analyze the possible nutritional factors leading to NCD and many other chronic diseases in this population. Based on our findings and according to the MNA evaluation, almost half of the examined Hungarian elderly are malnourished or at risk of malnutrition, and there is a significantly higher risk for women than for man. Overall, 23.5% of M responders and 31.5% of F reported weight loss, and among F there was a positive correlation between appetite loss and weight decrease. Among M, the modifying effect of appetite was even higher (70%). Malnutrition was widespread in the elderly and represented a major geriatric syndrome [31]. Maintaining an adequate nutritional status and a sufficient nutrient intake is key to health and good quality of life [32]. According to Vellas and colleagues (1999) [27], 22% of the elderly in a big sample were at risk of malnutrition or experiencing malnutrition already. In our study, MNA scores show that 7.4% of M and 16.2% of F are malnourished. Turkish older adults were more likely to be malnourished or at risk of malnutrition (4.9% and 31.5%) than Portuguese ones (1.2% and 24.0%) [7], but in each case this was at a lower rate than our responders. In Europe, low income is associated with food insecurity, with low levels of social protection. Food insecurity (limited availability of nutritionally adequate and safe food) is a potential risk factor for malnutrition [33]. Hungary is a LMIC; the consequences of this are manifold and are strongly correlated to physiological backfall and diseases [27]. A total MNA score below 24 shows malnutrition, with poor nutritional intake, but without weight loss, or low albumin level, while patients with MNA scores ≤17 probably have low albumin levels and experience weight loss [27]. The reliability of the MNA tool is very high, even in inter-observer relations. Studies show that a higher proportion of older subjects have dysphagia (swallowing problems). It is associated with depression, malnutrition, and with a lower proportion of subjects engaging in economic activity [34]. Patients with dysphagia were found to be highly dependent on meal behavior, and were mostly not able to prepare their own meal.

On the other hand, in our sample, the mean BMI values for M and F are 26.95 and 26.05, respectively, with 52.5% of subjects falling in the overweight range. The distribution of individuals with BMI higher than 25 shows that significant gender differences exist in the older age groups (50–55 and 56–60 years). In the USA, in the geriatric population (60 years and older), 37.1% of men and 33.6% of women are estimated to be overweight (BMI ≥ 30 kg/min^2^) [35,36]. Comparing malnutrition based on BMI data, 3.6% of F were malnourished, with MNA scores 16.2%. On the basis of BMI and its comparison to MNA scores, among females 26% of obese subjects were at the risk of malnutrition, while among overweight males the incidence was 84%. The difference was huge, showing that BMI calculation only increases the risk of data misinterpretation. The same fact is true if one accepts self-assessments. Most of the subjects reported no problems with eating or not remembering their own diets well. Although obesity is an important cause of chronic diseases, particularly in the elderly, one might declare that BMI is not a sufficient measure to assess health in older individuals and that other factors should be considered. However, in another Hungarian study, 84% of M and 76% of F participants were overweight or obese [37]. Body composition assessments show the ratio of muscle and fat, and this is an accurate but expensive and time-consuming measurement method. In our study, questionnaires and easy body measurements were applied. Analyzing dietary habits of the studied population, many elderly were found to be eating only calories without nutritional values, and their disadvantageous body composition change was demonstrated by comparing BMI to the level of malnutrition.

Lower-socioeconomic-status subjects tend to eat larger portions of unhealthy snacks, rather than small ones, leading to a potential 15–22% increase in energy intake [2]. In a Hungarian study socioeconomic differences in nutritional status were analyzed and “unhealthy diet” was found among 70.6% of Hungarians [38]. Although cooking with lard decreased dramatically, high-fat and CH-containing dishes are still popular among the elderly [22]. Subjects in our study were consuming a lot of CH. For females, this was on average 205 g, while for males it was 245 g, and it was 49% of daily caloric intake among F and 46.8% among M. Higher added sugar increased the risk of hypertension among the elderly as well. A 2.3-spoonful reduction in sugar amount decreased systolic blood pressure, while whole-fruit consumption was associated with the decrease in diastolic blood pressure [39]. In the Hungarian population, the ratio of hypertension is the highest in Europe [40]. The guidelines suggest that the free sugar content of food should not exceed 5% of daily calories [41]. Although the amount of added sugar was not calculated, the Hungarian diet is rich in sweetened foods, and pastry, pasta consumption. Many traditional dishes contain added sugar as a flavoring, intensifying the taste of vegetables. High CH intake decreases micronutrient consumption. In a prospective study, a 100 mL increase in sugar-sweetened beverages was associated with 40 mg lower calcium intake, with reduced iron and magnesium intake [42]. According to WHO guidelines, free sugar intake should be ˂10% of energy consumption. Others recommend that 25% of added sugars should be the upper limit of CH energy consumption [43]. In Australia, free sugar and micronutrient consumption oppositely are correlated, especially if free CH consumption is above the upper health limit. In our sample, the source of high CH intake was mostly white bread, pasta, potato, and rice. We also need to keep in mind that people living on lower incomes buy cheaper sweetened dairy products as well. According to the recommendations, 10–35% of the diet should come from proteins in the elderly in order to slow down sarcopenia [44]. Emotional eating was found to be caused by experiencing financial strain, rather than by traditional socioeconomic status dimensions in women, while restrained eating was associated with higher household income levels in women and with higher occupational position in men [45]. In Hungary, elderly people living on low incomes face challenges, including in terms of managing diet. Unhealthy, energy-dense foods are purchased more frequently in these populations [46]. The minimum protein consumption should be 30 g/day, although the elderly with acute or chronic diseases have higher protein demands (between 1.2–1.5 g/kg body weight/day) due to impaired anabolic responses to protein [18,47]. According to our findings, dietary protein intake is below the recommended daily allowance in 38% of M and 41% of F [44]. In our sample, both genders on average ate more protein than the minimum recommendation, but the ratio of protein in the total caloric intake was unfavorable (~15%) instead of the suggested recommendations of the easily digestible level of 30–35%. Similarly to earlier findings [22], males preferred meet, cold cuts, eggs and sausages, and females ate more dairy products than males. Diet is modifiable and a greater protein intake might lead to slower sarcopenia and a deterioration of health [47]. According to these data, among the elderly, the loss of lean body mass and the increase in fat mass can be modified by dietary caloric reduction and by increasing the protein amount of food above 25% of the daily energy intake [48]. Sarcopenic patients and non-sarcopenic elderly individuals consumed approximately the same amounts of protein on a daily basis, meeting the lowest level of recommendations, but after calculating the level of protein consumption by body weight, sarcopenic patients were found to consume significantly lower amounts of protein, vitamins B6, B12 and vitamin D compared to non-sarcopenic individuals, suggesting that a more protein-rich diet might lead to a higher amount of muscle and lower fall rate in the elderly [49]. In our sample, similarly to other Hungarian data, M were found to consume more protein than F, unlike in an elderly sample from Ecuador [50], in which F ate animal and plant proteins on a daily basis, while M did so only weekly. With aging, many meals are skipped due to a lack of appetite. In our sample, more than 70% of subjects had three meals within a day, but several responders had only two. For aging subjects, four or five light meals are suggested daily. Although most of the recommendations for elderly suggest 45–65% of daily CH intake, it was shown that a low level of CH consumption will decrease blood pressure, insulin level, and aging, and increase longevity [51]. In our population, although not refined sugar consumption, but the total amount of CH was very high, especially among F, compared to protein. Fat consumption of F was lower, but of M approximately at the border of metabolic syndrome and fat ratio [52]. In our study, obesity was more frequent among M, but the risk of malnutrition was higher in F. Most of the responder males reported their wife cooks for them, or they order food, while females, living alone on very low income will eat less food, but food that is cheap and high in CH. 

According to the World Health Organization [53], a minimum of 400 g or five portions of fruits and vegetables are suggested daily [54]. Despite these suggestions, the consumption of fruits and vegetables is still very low in the majority of countries [54]. In a 2019 survey conducted by the Hungarian National Statistics Agency in Hungary [55], 75% of people aged 65 years and over consumed fruit daily, and more than half consumed vegetables. In our study, 72.3% of the elderly people asked consumed fruits and vegetables (66.8%) once a day. Research in China found (average age was around 85 years) that approximately 56% of respondents consumed fresh vegetables daily, and only 14% consumed fresh fruit daily [56]. In another Chinese study asking elderly people (over 60 years of age), >60% of the participants consumed ≥2 servings of fruit,  >70% consumed ≥5 servings of vegetables on an average day [57], and women had higher fruit and vegetable intake than men. Conversely, in Uganda, only 12% of the population consumed five or more servings of fruits and vegetables daily [58]. In Spain (mean age 74.8 years), fruit and vegetable intake amounted to four servings per day on average. Participants consuming less were more likely to be men [17]. A cross-sectional study in the Netherlands concluded that physically active older adults tended to consume more fruits and vegetables compared to their less active peers [59]. After reviewing published data, Mayen et al. (2014) [4] found that lower fruit and vegetable consumption were reported in LMICs compared to the high-income countries, having lower quality diet. This meant that, while energy intake was not significantly different, a decrease in CH and increase in protein and fat consumption were measured in HICs. Interestingly, fruit intake in LMICs was lower in urban areas compared to the rural ones [4], vegetable consumption was related only to behavioral determinants, and fruit consumption was influenced only by economic status [60]. Lower income and retirement are associated with increased odds of impaired fasting glucose and type 2 diabetes in China. Education and occupation might play a role in glycemic control among patients with type 2 diabetes [61]. Inflammatory diseases, such as irritable bowel disease, are also associated with an imbalance in the gut microflora. There are many data suggesting that dietary factors play a significant role in the onset and progression of the disease, affecting the gut microbial flora’s composition and its function [62]. Processed foods, refined sugar, and fat will decrease the microbiome of the gut, increasing the development of immunological diseases.

A study in Europe investigated the fluid intake of elderly subjects. Among females, water intake was below the cut-off value in a high percentage of cases. They were found to be at higher risk of dehydration than men because of much lower water intakes and the overall relationship between fluid intake and a poor mental state and ADL problems [63]. In a Polish study, the average water intake in elderly women met the recommendations of the European Food Safety Agency, but men did not reach it (about 200 mL less daily) [64]. A nationwide, representative German study showed that 14% of all participants (men 15%, women 14%) reported usual drinking amounts below 1 L, and this ratio was markedly higher in subjects above 85 years (27%) compared to younger participants (75–84 y, 15%, 65–74 y, 8%; *p* = 0.001) [65]. In our study, significant differences were found between genders in water consumption (F liquid consumption was higher than that of M *p* ≤ 0.01.), but the general water intake among the elderlies was less than the recommendations.

Only 27.93% of the respondents took dietary supplements, which could be increased with health education. This is similar to earlier findings, in which almost one-third of the elderly took some kind of vitamins and/or mineral supplements [22]. The amount of respondents consuming fruits (11.7%) and vegetables (8.93%) several times a day was extremely low, which is unfavorable, since eating vegetables and fruits would cover the necessary dietary fiber, mineral, and vitamin needs in this age group [66]. To counterbalance this, the use of nutritional supplements may be recommended, allowing patients to avoid chronic diseases resulting from inadequate nutrient intake [18] or muscle loss, which can be influenced by vitamins and antioxidants [47]. Daily multivitamin intake significantly reduces the risk of developing cancer by 8% [67]. To counterbalance bone loss, the consumption of calcium and vitamin D3 may be recommended [68]. Jackson and colleagues (2006) [69] showed that the use of calcium in combination with vitamin D3 significantly increased hip bone density, but could not significantly reduce the frequency of bone fractures. In postmenopausal females, lean mass was the best determinant of bone mineral density [8]. Lean body mass, years since menopause, and dietary patterns are the important determinants of this in the spine, hip, and the total body. A positive correlation was found between α-klotho and trunk bone mineral density, showing its value for predicting osteoporosis. The significant effect of α-klotho (factor of bone metabolism) on bone mineral density in diabetes patients suggests its potential as a predictive marker of diabetes progression [70], important in an aging society, in addition to bone metabolism.

A 2-year multilevel intervention was conducted on energy balance-related behaviors among European families at risk of developing type 2 diabetes, based on self-reported physical activity, sedentariness, and eating behaviors in Belgium, Finland, Greece, Spain, Hungary, and Bulgaria [71]. Unfavorable intervention effects were found due to the consumption of soft drinks and sugar-containing juices among Hungarian children and parents. Examining the intervention effects for the overall population and per country, we found that 10 of the 112 investigated outcome variables were improved in the intervention group compared to the controls. In addition to diet, the activity level of the elderly led to healthier aging. Daily step counts were associated with all-cause mortality in patients with congestive heart failure, and 5581 daily steps were associated with a decreased risk of all-cause mortality in patients [72], although, in women there was no difference in mortality for each 1000 steps/day increase. In fact, no significant difference was found between less than 5581 and more steps among females. In a Hungarian study of adults (mean age was 44.41 ± 18.64 years), the average number of daily steps was found to be relatively high 7308.47 [73]. According to a Canadian study 7100 steps/day would be approximately enough for a healthy elderly person (7000–10,000) [74].

Aging is also associated with a decrease in gastrointestinal functions, leading to lower abilities to generate protective immunity, and increased incidence of inflammation and oxidative stress. So, gastrointestinal disorders occur more frequently in the elderly population [75]. Dietary choices have an impact on the microbial flora of the gut in people with inflammatory diseases [76], and so the relationship between diet, gut health, and the risk of developing obesity, cardiovascular, and inflammatory diseases is important. A high-fat diet is also related to atherogenesis and a study identified dysregulated lncRNAs and mRNAs that may be potential biomarkers or drug targets of blood vessel inflammations [77].

Further special analysis and research (nutritional factors and Mini Nutritional Assessment) should explore the specific health implications and reasons behind our findings.

The limitations of our study are partly due to the low number of volunteers responding to the study. One cannot reach the general adult Hungarian population in a high number, especially not among the elderly. Although the educational level might strongly modify the obtained data, in Hungary most of the population learn about healthy food and diet. Although during data collection several factors were considered in order to avoid data misinterpretation, one must keep in mind that socioeconomic status limits the quality of food consumed. Indeed, overreporting might shade the results obtained.

## 6. Conclusions

The nutritional habits of the Hungarian elderly, retired population, similarly to other data, are strongly correlated to the socioeconomic status, original educational level, and lifestyle habits of persons [2,4,38]. There are gender differences in the possibility of malnutrition (in favor of females), which correlates with the loss of appetite and lower meal consumption. Males have higher values of BMI, with more cases of overweight or obesity [55]. The difference in food product consumption partly correlates with the fact that M eat more often out of the home, where meals contain much more vegetables and fruits than homemade food. Changes in the socioeconomic status of this population during the last decades are visible in the increases in amounts consumed, changing from less protein and more vegetables in the diet to the opposite. The increase in CH consumption is measurable in the type of food eaten as well, marked by more white bread, potatoes, and pasta. In the traditional Hungarian diet, as in many LMICs, legumes are very important sources of CH and proteins, further increasing the amount of energy consumed. Water consumption is generally very low, which is in accordance with other data. One-third of the studied population drinks less than 1 L, and 15% of responders drink only half a liter water daily, although M drink significantly more. Although a very high percentage of Hungarian elderly have two or three chronic diseases, visiting medical centers and meeting professionals regularly, their general knowledge of a healthy diet is still low. Changes in education showing the advantages of a healthy lifestyle, even in lower-income families, might help to improve the general health and expected lifespan of Hungarians. In the case of a failure to change the regression of healthy habits and lifestyle, more diseases connected to immunosenescence and digestive system problems will be measured in this aging society.

## Figures and Tables

**Figure 1 nutrients-16-01203-f001:**
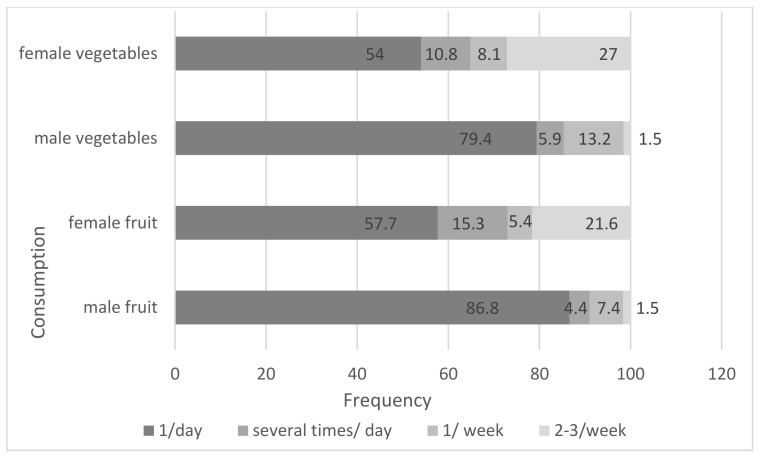
Frequency of fruit and vegetable consumption of the studied elderly population.

**Table 1 nutrients-16-01203-t001:** Evaluation based on MNA total score.

	Males (*n* = 68) %	Females (*n* = 111) %	Chi^2^ Test *p*
malnourished	7.4	16.2	n.s
risk of malnutrition	14.7	40.5	<0.001

n.s not signifficant.

**Table 2 nutrients-16-01203-t002:** Self-evaluation of subjects based on MNA self-assessment.

	Males (*n* = 68) %	Females (*n* = 111) %	Chi^2^ Test *p*
malnourished	2.9	1.8	n.s
uncertain in own assessment	17.6	36.9	0.006
no problem with eating	79.4	61.2	0.011

n.s not signifficant.

**Table 3 nutrients-16-01203-t003:** Average BMI of the responders.

BMI (kg/m^2^)	*n*	Min	Max	Mean ± SD
Males	68	21.46	32.11	26.95 ± 2.35
Females	111	17.04	37.75	26.05 ± 4.43

**Table 4 nutrients-16-01203-t004:** Distribution of the population having BMI > 25.

Age (Year)	Males (*n* = 68) %	Females (*n* = 111) %	Chi^2^ Test *p*
50–55	70	36.4	0.044
56–60	90	40	0.012
61–65	100	73.3	n.s
66–70	60	73.3	n.s
71–75	80	80	n.s
76–80	83.3	66.7	n.s
81–85	50	62.5	n.s
85<	66.7	33.3	n.s

n.s not signifficant.

**Table 5 nutrients-16-01203-t005:** Protein, carbohydrate and fat consumption of males and females in age groups.

Age (Year)	Males (*n* = 68)				Females (*n* = 111)			
	Protein (g)	CH (g)	Fat (g)	KCal	Protein (g)	CH (g)	Fat (g)	KCal
50–55	81	247	89	2148	61	195	57	1598
56–60	81	288	93	2302	65	190	65	1690
61–65	97	275	92	2350	54	194	60	1632
66–70	81	236	85	2152	66	223	66	1792
71–75	75	273	87	2224	76	233	69	1912
76–80	67	198	69	1698	62	213	69	1677
81–85	78	196	90	1858	63	221	53	1669
85>	69	250	75	1996	57	177	47	1367
Mean ± SD	78 ± 9.2	245 ± 34.3	85 ± 8.6	2091 ± 224.9	63 ± 6.6	205 ± 19.5	60 ± 8	1667 ± 156.9

**Table 6 nutrients-16-01203-t006:** Frequency of fruit and vegetable consumption in the studied population.

Frequency of Fruits Consumption	Males (*n* = 68) %	Females (*n* = 111) %	Chi^2^ Test *p*
Once a day	86.8	57.7	<0.001
Several times a day	4.4	15.3	0.025
Once a week	7.4	5.4	n.s
2–3 times a week	1.5	21.6	<0.001
Frequency of vegetables consumption			
Once a day	79.4	54.1	0.001
Several times a day	5.9	10.8	n.s
Once a week	13.2	8.1	n.s
2–3 times a week	1.5	27	<0.001

n.s not signifficant.

**Table 7 nutrients-16-01203-t007:** Frequency of taking dietary supplements in the sample.

Age (Year)	Males (*n* = 68)		Females (*n* = 111)	
	Do you Take Dietary Supplements?			
	Yes	No	Yes	No
50–55	1	9	8	14
56–60	2	8	5	10
61–65	1	9	6	9
66–70	1	9	4	11
71–75	0	10	6	9
76–80	1	5	7	8
81–85	2	4	5	3
85<	0	6	1	5
Total	8	60	42	69
%	11.76	88.23	37.83	62.16
Mean ± SD	1 ± 0.75	5 ± 2.2	5.25 ± 2.12	8.62 ± 3.42

n.s not signifficant.

**Table 8 nutrients-16-01203-t008:** Water consumption of the asked elderly population.

Liquid Consumption per Day	Males (*n* = 68) %	Females (*n* = 111) %	Chi^2^ Test *p*
0.5 L	31	5.4	*p* < 0.001
1 L	26.5	42.3	*p* = 0.032
2 L	42.5	33.3	n.s
above 2 L	0	19	*p* < 0.001

## Data Availability

Data is unavailable due to ethical restrictions, but data can be made available upon reasonable request and ethical approval.

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
