# Peer review of "Nutritional Habits of Hungarian Older Adults"

_nutrients, 2024, doi:10.3390/nu16081203_

Round 1

Reviewer 1 Report (Previous Reviewer 1)

Comments and Suggestions for Authors

The resubmitted manuscript entitled Eating Habits of Hungarian Older Adults by Rita et al in which the authors aimed to analyze the feeding habits of a Hungarian elderly population, energy and nutrient intake, focusing on macronutrients, water, fruit and vegetable consumption searching possible nutritional factors leading to NCD, and many other chronic diseases in this population. The manuscript is based on a very good idea but it needs improvement before it can be accepted for publication.

The authors must revise the manuscript according to the following comments.

Introduction

The title indicates the eating habits but in the whole introduction, I didn’t see a detailed paragraph about eating habits neither or types of diets.

The authors must add the latest literature on the eating habits and various diets consumed by elderly people in Hungary.

In addition to that the authors must also include a paragraph on several foods which are linked to aging and other inflammatory diseases.

The following articles provided comprehensive details which has been published recently in Nutrinets

Elucidating the role of diet in maintaining gut health to reduce the risk of obesity, cardiovascular and other age-related inflammatory diseases: recent challenges and future recommendations. Gut Microbes. 2024 Jan-Dec;16(1):2297864. doi: 10.1080/19490976.2023.2297864.

Dietary Implications of the Bidirectional Relationship between the Gut Microflora and Inflammatory Diseases with special emphasis on Irritable Bowel Disease: Current and Future Perspective. Nutrients 2023, 15, 2956. https://doi.org/10.3390/nu15132956

Zhou, Y., Sun, X., Yang, G., Ding, N., Pan, X., Zhong, A.,... Chai, X. (2023). Sex-specific differences in the association between steps per day and all-cause mortality among a cohort of adult patients from the United States with congestive heart failure. Heart & Lung, 62, 175-179. doi: https://doi.org/10.1016/j.hrtlng.2023.07.009

Zhang, Y., Zhao, C., Zhang, H., Chen, M., Meng, Y., Pan, Y., Zhuang, Q., & Zhao, M. (2023). Association between serum soluble α-klotho and bone mineral density (BMD) in middle-aged and older adults in the United States: a population-based cross-sectional study. Aging clinical and experimental research, 35(10), 2039–2049. https://doi.org/10.1007/s40520-023-02483-y

Bao, M., Luo, H., Chen, L., Tang, L., Ma, K., Xiang, J.,... Li, J. (2016). Impact of high fat diet on long non-coding RNAs and messenger RNAs expression in the aortas of ApoE(−/−) mice. Scientific Reports, 6(1), 34161. doi: 10.1038/srep34161

Chen, Y., Xiang, J., Wang, Z., Xiao, Y., Zhang, D., Chen, X.,... Zhang, Q. (2015). Associations of Bone Mineral Density with Lean Mass, Fat Mass, and Dietary Patterns in Postmenopausal Chinese Women: A 2-Year Prospective Study. PLOS ONE, 10(9), e137097. doi: 10.1371/journal.pone.0137097

The authors should also add some details about exercise as well.

Please add some latest literature / references to discussion section.

Figures quality should be improved.

Add a paragraph about future perspective as well.

Author Response

The resubmitted manuscript entitled Eating Habits of Hungarian Older Adults by Rita et al in which the authors aimed to analyze the feeding habits of a Hungarian elderly population, energy and nutrient intake, focusing on macronutrients, water, fruit and vegetable consumption searching possible nutritional factors leading to NCD, and many other chronic diseases in this population. The manuscript is based on a very good idea but it needs improvement before it can be accepted for publication.

The authors must revise the manuscript according to the following comments.

Introduction

The title indicates the eating habits but in the whole introduction, I didn’t see a detailed paragraph about eating habits neither or types of diets.

Thank you very much for your comment! Indeed the title might be misleading, so we have changed it to: Nutritional habits of Hungarian older adults. We hope this title might serve the intention of authors better.

The authors must add the latest literature on the eating habits and various diets consumed by elderly people in Hungary.

Thank you very much for the suggestion! The following paragraph describing Hungarian findings was added to introduction:

In the Hungarian population neither nutrition, nor the physical activity status seems to meet the European standards. Füzesi et al (2004) [21] described in a large scale study with the aid of SF36 that the age group between 55-64 years has very low scores in overall health, physical functions, physical role, pain and vitality. The measured population reported very low levels of daily physical activities (35%, or more doing nothing, not even walking). Considering BMI categories most of the overweight and obese population were older, having the lowest general health and physical functions. Subjects in the underweight category were much younger (30.82y average), having the scores in general health, vitality and physical function, as in the normal BMI range. In a dietary habits and activity measuring survey of Hungarian elderly [22] nearly 60% of people had three meals per day, which increased with ageing. The most substantial meal was lunch, meals were prepared at home by 90% of the elderly. Lard for cooking was used by 44% of subjects. Milk, dairy products, fish, fruits, fresh vegetables and vegetable dishes were consumed far below the recommendations. All types of meat were radically decreased, especially beef compared to former official Hungarian surveys before. Legumes were consumed several times within a week. The average time spent with outdoor activities was 12 h per week, consisting of walking, gardening, shopping, but not from exercises. According to official statistical data, the quality of life index was found to be quite low in Hungary (2023-2024 data) compared to other European countries, low purchasing power (49.8) and high property price ratio to income was reported (Quality of Life Index by Country 2024 (numbeo.com)).Within Hungary the South-Danubean region is having even lower incomes. Unfavorable changes compared to a nationwide previous survey (from 2009 to 2014) were seen, the increase of fat and saturated fatty acid energy percent in women, the decrease in fruit and vegetable consumption, explaining the decreased fiber intake. An increasing trend in added sugar energy percent in each age groups of both genders was observed compared to 2009, meanwhile the frequency of having meals also decreased (Sarkadi Nagy et al 2014). Similar trend in fat consumption was found in another national survey, mentioning the increased fat amount and higher cholesterol levels (Zámbó et al 2021).

Many other small, but important information was included in the Discussion section.

In addition to that the authors must also include a paragraph on several foods which are linked to aging and other inflammatory diseases.

The following articles provided comprehensive details which has been published recently in Nutrinets

Elucidating the role of diet in maintaining gut health to reduce the risk of obesity, cardiovascular and other age-related inflammatory diseases: recent challenges and future recommendations. Gut Microbes. 2024 Jan-Dec;16(1):2297864. doi: 10.1080/19490976.2023.2297864.

Dietary Implications of the Bidirectional Relationship between the Gut Microflora and Inflammatory Diseases with special emphasis on Irritable Bowel Disease: Current and Future Perspective. Nutrients 2023, 15, 2956. https://doi.org/10.3390/nu15132956

Zhou, Y., Sun, X., Yang, G., Ding, N., Pan, X., Zhong, A.,... Chai, X. (2023). Sex-specific differences in the association between steps per day and all-cause mortality among a cohort of adult patients from the United States with congestive heart failure. Heart & Lung, 62, 175-179. doi: https://doi.org/10.1016/j.hrtlng.2023.07.009

Zhang, Y., Zhao, C., Zhang, H., Chen, M., Meng, Y., Pan, Y., Zhuang, Q., & Zhao, M. (2023). Association between serum soluble α-klotho and bone mineral density (BMD) in middle-aged and older adults in the United States: a population-based cross-sectional study. Aging clinical and experimental research, 35(10), 2039–2049. https://doi.org/10.1007/s40520-023-02483-y

Bao, M., Luo, H., Chen, L., Tang, L., Ma, K., Xiang, J.,... Li, J. (2016). Impact of high fat diet on long non-coding RNAs and messenger RNAs expression in the aortas of ApoE(−/−) mice. Scientific Reports, 6(1), 34161. doi: 10.1038/srep34161

Chen, Y., Xiang, J., Wang, Z., Xiao, Y., Zhang, D., Chen, X.,... Zhang, Q. (2015). Associations of Bone Mineral Density with Lean Mass, Fat Mass, and Dietary Patterns in Postmenopausal Chinese Women: A 2-Year Prospective Study. PLOS ONE, 10(9), e137097. doi: 10.1371/journal.pone.0137097

Thank you very much for the suggestions! Although some of the manuscripts were already mentioned in our paper, now we included them in more details. Some other manuscripts, not mentioned by the reviewer were also addedto the text and references.

The authors should also add some details about exercise as well.

Indeed physical activity is strongly correlated to health parameters. The following Hungarian study was added to Introduction: In a dietary habits and activity measuring survey of Hungarian elderly [22] nearly 60% of people had three meals per day, which increased with ageing. The most substantial meal was lunch, meals were prepared at home by 90% of the elderly. Lard for cooking was used by 44% of subjects. Milk, dairy products, fish, fruits, fresh vegetables and vegetable dishes were consumed far below the recommendations. All types of meat were radically decreased, especially beef compared to former official Hungarian surveys before. Legumes were consumed several times within a week. The average time spent with outdoor activities was 12 h per week, consisting of walking, gardening, shopping, but not from exercises

Discussion:

A 2-year multilevel intervention, on energy balance-related behaviors among European families at risk for developing type 2 diabetes was conducted, based on self-reported physical activity, sedentary and eating behaviors in: Belgium, Finland, Greece, Spain, Hungary, and Bulgaria [67]. Unfavorable intervention effects were found on the consumption of soft drinks and sugar-containing juices among Hungarian children and parents, while examining the intervention effects for the overall population and per country, 10 from the 112 investigated outcome variables were improved in the intervention group compared to the controls. Besides diet the activity level of elderly leads to healthier ageing. Daily step counts are associated with all-cause mortality in patients with congestive heart failure, 5581 daily steps were associated with a decreased risk of all-cause mortality in patients [68], although, in women there was no difference on mortality for each 1000 steps/day increase. In fact, no significant difference was found between less than 5581 and more steps among females. In a Hungarian study of adults (mean age was 44.41±18.64 years) the average number of daily steps was found to be relatively high 7308.47 [69]. According to a Canadian study 7100 steps/day would be approximately enough for healthy elderly (7000-10000) [70].

Please add some latest literature / references to discussion section.

Thank you very much for your suggestion! Many articles are newly included in the manuscript, hoping, we have chosen the adequate ones. Right now 77 items are on the list of references.

Figures quality should be improved.

Thank you very much for your suggestion! The figure was newly created. The authors hope that it is more informative and its quality is much better!

Add a paragraph about future perspective as well.

Thank you very much for the suggestion! We did not include a whole paragraph, but th end of Conclusions was modified, hpoefully fulfilling the needs of possible future.

Although very high percentage of Hungarian elderly have two, or three chronic diseases, visiting medical centers, and meeting professionals regularly, their general knowledge of healthy diet is still low. Changes in education showing the advantages of a healthy lifestyle even in lower income families might help improving the general health and expected lifespan of Hungarians.

Finally, we greatly appreciate your detailed work and feedback on our manuscript! We hope that our revised manuscript will reach the standard of Nutrition Journal and will be satisfactory for publication!

Reviewer 2 Report (Previous Reviewer 2)

Comments and Suggestions for Authors

The study sample includes a higher proportion of females (62%) compared to males (38%), which may affect the generalizability of the findings, especially regarding gender-specific nutritional patterns.

The study focuses on energy and nutrient intake, fruit and vegetable consumption, and malnutrition risk. Other factors that could influence nutritional status, such as socio-economic status, physical activity level, and access to healthcare, are not addressed.

The data collected through questionnaires and feeding diaries rely on self-reporting, which can be subject to recall bias and misreporting. Participants may overestimate or underestimate their food intake or may not accurately report their nutritional habits considering the age criteria of the study participants.

Page 5, paragraph below table 2, line 4: The message is not clear, suggest revising.

Which information is correct, Based on MNA score: data mentioned in abstract or page 5 paragraph below table 2 and 3.

Data is contradicting and not clear.

Overall data representation is little clumsy and need revision.

Author Response

The study sample includes a higher proportion of females (62%) compared to males (38%), which may affect the generalizability of the findings, especially regarding gender-specific nutritional patterns.

Thank you very much for your note! Indeed, the ratio of sexes is different. The reason for that is included already in Materials and Methods:

Finally, 179 responses were analyzed (females (F): 111, males (M): 68). The ratio of genders is different, since males (similarly to other Hungarian studies) were not cooperative in this study either. 

Unfortunately we were working with volunteers, and males are nit cooperative in these kind of studies. Although data misinterpretation might be the consequence of that, but according to earlier findings, probably are data are very close to the results of national surveys.

The study focuses on energy and nutrient intake, fruit and vegetable consumption, and malnutrition risk. Other factors that could influence nutritional status, such as socio-economic status, physical activity level, and access to healthcare, are not addressed.

Indeed, the socioeconomic status was not asked, it was not included in the questionnaire, but as we describe in many details, the possibilities of finding proper nutrition is strongly related to the socio-economic status of Hungary.

Analyzing dietary habits of the studied population, many elderly were eating only calories without nutritional values, their disadvantageous body composition change is demonstrated comparing BMI to malnutrition.

Lower socioeconomic status subjects intend to eat larger portions of unhealthy snacks, rather than small ones, leading to a potential 15-22% increase in energy intake [2]. In a Hungarian study socioeconomic differences in nutritional status was analyzed and "unhealthy diet" was 70.6% among Hungarians [34]. Although cooking with lard decreased dramatically, high fat and CH containing dishes are still popular among elderly [22]. Subjects in our study were consuming a lot of CH, females on average 205g, while males 245g, and it is 49% of daily caloric intake among F and 46.8% among M.

In Hungary elderly living from low income face challenges, including managing diet. Unhealthy, energy-dense foods are purchased more frequently in these populations [42].

After reviewing published data Mayen et al (2014) [4] have found that lower fruit and vegetable consumption was reported in the LMIC, compared to the high-income countries, having lower quality diet, meaning that while energy intake was not significantly different, decrease in CH and increase in protein, fat consumption was measured in the HIC. Interestingly, in LMIC fruit intake was lower in urban areas compared to the rural ones [4], vegetable consumption was related only to behavioral determinants, fruit consumption was influenced only by economic status [56].

The data collected through questionnaires and feeding diaries rely on self-reporting, which can be subject to recall bias and misreporting. Participants may overestimate or underestimate their food intake or may not accurately report their nutritional habits considering the age criteria of the study participants.

 Data collection happened with the intention of collecting higher number of responders with useful and trustful results, so helpers were trained (nurses and physiotherapists, N=10) to ask and code the answers the same way, using the snowball method. Trained helpers were also explaining personally all the important questions, facts of the questionnaires, describing the possible answers in detail, helping to fill them correctly and trustfully.

The questions asked were matched to Hungarian know food types, helping to identify them, keeping in mind to possible wrong answers would help misinterpretation, the frequency questionnaire matched the standards mentioned by Bailey (2021).

The third part of this survey contained a 3-day diet diary. Two, not consecutive weekdays, and one weekend day was requested to track each day [25]. All the subjects had a training before the start of data collection, teaching the strategy of categorization. Subjects had to note each meal or drink, snack etc. separately in each day’s preformed sheet, describing the type and amount of food or drink. The weight and the type of the food (if prepacked, or sliced) were also asked, just like the number of cooked meals (1, or 2 plates). In case of liquid consumption it was expected of the participants to give their answers in mL, or cups.

According to other findings even the 24 hour recall method of diet analyzation can be very useful, especially if it is repeated, but not with consecutive days. We followed these recommendations, and also trained helpers and subjects. Although this way of data collection is depending on the memories of subjects, we have asked volunteers to fill the table immediately after aeting/drinking something.

Page 5, paragraph below table 2, line 4: The message is not clear, suggest revising.

Thank you, sentences were modified, hoping for better understanding.

Which information is correct, Based on MNA score: data mentioned in abstract or page 5 paragraph below table 2 and 3.

Data is contradicting and not clear.

 As we describe in the results, and also in the abstract, within the same population, but with two different tool (measurements), very different populations were identified, showing that BMI is not a good indicator in this population.

The BMI categories of subjects were diverse. According to measurements within the same population the following data were collected (Table 3.). The ratio of having BMI˃25 was much higher among M in each age group, significant difference was found in the 50-60 years population (Table 4). Interestingly enough in the 61-65 years age group 100% of M, and 73% of F were in the overweight category. Surprisingly among females 59% was in the overweight and obese category after BMI calculation, but analyzing the same population according to the MNA scores 26% (17 females) of them fell into the category of risk of malnutrition. Among males 76.3% was overweight or obese, while 9.6% (5 subjects) of them fell into the risk of malnutrition based on MNA scores. In the whole sample 117 subjects (65%) were overweight or obese and 18.8% (22) of them were in the risk of malnutrition.

Hopefully description of data and the new figure helps understanding the results in a more compact way.

Overall data representation is little clumsy and need revision.

Thank you very much for your comment!

The figure was newly created. The authors hope that it is more informative and its quality is much better!

 We have included several facts, details, so the Discussion is more detailed, but we hope it helps the understanding and interpretation of our findings.

Finally, we greatly appreciate your detailed work and feedback on our manuscript! We hope that our revised manuscript will reach the standard of Nutrition Journal and will be satisfactory for publication!

Round 2

Reviewer 1 Report (Previous Reviewer 1)

Comments and Suggestions for Authors

The authors have revised the manuscript and can be accepted for publication now

Reviewer 2 Report (Previous Reviewer 2)

Comments and Suggestions for Authors

I appreciate authors for addressing all the comments and revising the manuscript.

I have no further comments for this manuscript.

This manuscript is a resubmission of an earlier submission. The following is a list of the peer review reports and author responses from that submission.

Round 1

Reviewer 1 Report

Comments and Suggestions for Authors

The manuscript entitled Eating Habits of Hungarian Older Adults by Rita et al in which the authors aimed to analyze the feeding habits of a Hungarian elderly population, energy and nutrient intake, focusing on macronutrients, water, fruit and vegetable consumption searching possible nutritional factors leading to NCD, and many other chronic diseases in this population. The manuscript is based on a very good idea but it needs improvement before it can be accepted for publication.

The authors must revise the manuscript according to the following comments.

Introduction

The title indicates the eating habits but in the whole introduction, I didn’t see a detailed paragraph about eating habits neither or types of diets.

The authors must add the latest literature on the eating habits and various diets consumed by elderly people in Hungary.

In addition to that the authors must also include a paragraph on several foods which are linked to aging and other inflammatory diseases.

The following articles provided comprehensive details which has been published recently in Nutrinets

Dietary Implications of the Bidirectional Relationship between the Gut Microflora and Inflammatory Diseases with special emphasis on Irritable Bowel Disease: Current and Future Perspective. Nutrients 2023, 15, 2956. https://doi.org/10.3390/nu15132956

Zhou, Y., Sun, X., Yang, G., Ding, N., Pan, X., Zhong, A.,... Chai, X. (2023). Sex-specific differences in the association between steps per day and all-cause mortality among a cohort of adult patients from the United States with congestive heart failure. Heart & Lung, 62, 175-179. doi: https://doi.org/10.1016/j.hrtlng.2023.07.009

Zhang, Y., Zhao, C., Zhang, H., Chen, M., Meng, Y., Pan, Y., Zhuang, Q., & Zhao, M. (2023). Association between serum soluble α-klotho and bone mineral density (BMD) in middle-aged and older adults in the United States: a population-based cross-sectional study. Aging clinical and experimental research, 35(10), 2039–2049. https://doi.org/10.1007/s40520-023-02483-y

Bao, M., Luo, H., Chen, L., Tang, L., Ma, K., Xiang, J.,... Li, J. (2016). Impact of high fat diet on long non-coding RNAs and messenger RNAs expression in the aortas of ApoE(−/−) mice. Scientific Reports, 6(1), 34161. doi: 10.1038/srep34161

Chen, Y., Xiang, J., Wang, Z., Xiao, Y., Zhang, D., Chen, X.,... Zhang, Q. (2015). Associations of Bone Mineral Density with Lean Mass, Fat Mass, and Dietary Patterns in Postmenopausal Chinese Women: A 2-Year Prospective Study. PLOS ONE, 10(9), e137097. doi: 10.1371/journal.pone.0137097

The authors should also add some details about exercise as well.

Please add some latest literature / references to your article.

Figures quality should be improved.

Conclusion should be added to this study.

Comments on the Quality of English Language

The manuscript entitled Eating Habits of Hungarian Older Adults by Rita et al in which the authors aimed to analyze the feeding habits of a Hungarian elderly population, energy and nutrient intake, focusing on macronutrients, water, fruit and vegetable consumption searching possible nutritional factors leading to NCD, and many other chronic diseases in this population. The manuscript is based on a very good idea but it needs improvement before it can be accepted for publication.

The authors must revise the manuscript according to the following comments.

Introduction

The title indicates the eating habits but in the whole introduction, I didn’t see a detailed paragraph about eating habits neither or types of diets.

The authors must add the latest literature on the eating habits and various diets consumed by elderly people in Hungary.

In addition to that the authors must also include a paragraph on several foods which are linked to aging and other inflammatory diseases.

The following articles provided comprehensive details which has been published recently in Nutrinets

Dietary Implications of the Bidirectional Relationship between the Gut Microflora and Inflammatory Diseases with special emphasis on Irritable Bowel Disease: Current and Future Perspective. Nutrients 2023, 15, 2956. https://doi.org/10.3390/nu15132956

Zhou, Y., Sun, X., Yang, G., Ding, N., Pan, X., Zhong, A.,... Chai, X. (2023). Sex-specific differences in the association between steps per day and all-cause mortality among a cohort of adult patients from the United States with congestive heart failure. Heart & Lung, 62, 175-179. doi: https://doi.org/10.1016/j.hrtlng.2023.07.009

Zhang, Y., Zhao, C., Zhang, H., Chen, M., Meng, Y., Pan, Y., Zhuang, Q., & Zhao, M. (2023). Association between serum soluble α-klotho and bone mineral density (BMD) in middle-aged and older adults in the United States: a population-based cross-sectional study. Aging clinical and experimental research, 35(10), 2039–2049. https://doi.org/10.1007/s40520-023-02483-y

Bao, M., Luo, H., Chen, L., Tang, L., Ma, K., Xiang, J.,... Li, J. (2016). Impact of high fat diet on long non-coding RNAs and messenger RNAs expression in the aortas of ApoE(−/−) mice. Scientific Reports, 6(1), 34161. doi: 10.1038/srep34161

Chen, Y., Xiang, J., Wang, Z., Xiao, Y., Zhang, D., Chen, X.,... Zhang, Q. (2015). Associations of Bone Mineral Density with Lean Mass, Fat Mass, and Dietary Patterns in Postmenopausal Chinese Women: A 2-Year Prospective Study. PLOS ONE, 10(9), e137097. doi: 10.1371/journal.pone.0137097

The authors should also add some details about exercise as well.

Please add some latest literature / references to your article.

Figures quality should be improved.

Conclusion should be added to this study.

Author Response

Dear Reviewer,

Thank you very much for your work your very helpful feedback a constructive suggestions!

Indeed with your feedback and suggestions we have the opportunity to improve our manuscript Below are the actions, comments/thoughts we have taken to improve the quality of the manuscript:

The manuscript entitled Eating Habits of Hungarian Older Adults by Rita et al in which the authors aimed to analyze the feeding habits of a Hungarian elderly population, energy and nutrient intake, focusing on macronutrients, water, fruit and vegetable consumption searching possible nutritional factors leading to NCD, and many other chronic diseases in this population. The manuscript is based on a very good idea but it needs improvement before it can be accepted for publication.

The authors must revise the manuscript according to the following comments.

Introduction

The title indicates the eating habits but in the whole introduction, I didn’t see a detailed paragraph about eating habits neither or types of diets.

Thank you very much for your comment! Indeed the title might be misleading, so we have changed it to: Nutritional habits of Hungarian older adults. We hope this title might serve the intention of authors better.

The authors must add the latest literature on the eating habits and various diets consumed by elderly people in Hungary.

The following important facts were included in our work:

Higher sugar intake will increase calorie intake, the probability of overweight, obesity and development of noncommunicable diseases [1], just like socioeconomic status being one of the strongest predictors of obesity and unhealthy food environments [2]. Free sugar consumption and NCD are higher in the low- or middle-income countries (LMIC), like Hungary

There are many data discussing socioeconomic determinants in dietary patterns [4], in LMIC there was a strong correlation between fruit and vegetable, just like healthy food consumption and income of the population.

Many other important details are included in Discussion.

In addition to that the authors must also include a paragraph on several foods which are linked to aging and other inflammatory diseases.

The reviewer is perfectly right in this fact, the suggestion is very important! Since there are many diseases mentioned in the introduction and the space is limited in a manuscript, we have included the following sentences in Introduction:

Osteopenia and osteoporosis are also important consequences of aging, and so are unfavorable lifestyle habits. In a Chinese postmenopausal population lean mass was the best determinant of bone mineral density [7]. Six dietary patterns were also identified but only cereal grains-fruits and milk-root vegetables patterns were associated with bone density of the spine and hip.

In the Discussion:

Lower socioeconomic status subjects intend to eat larger portions of unhealthy snacks, rather than small ones, leading to a potential 15-22% increase in energy intake (Best et al, 2019). In a Hungarian study socioeconomic differences in nutritional status was analyzed and "unhealthy diet" was 70.6% among Hungarians [29]. Emotional eating was found to be rather caused by experiencing financial strain, rather than by traditional socioeconomic status dimensions in women, while restrained eating was associated with higher household income level in women and with higher occupational position in men [33]. In Hungary elderly living from low income face challenges, including managing diet. Unhealthy, energy-dense foods are purchased more frequently in these populations [34].

The following articles provided comprehensive details which has been published recently in Nutrients

Thank you very much for your detailed help! Indeed data and manuscripts mentioned below are very important and new! Only because of considering the length of the manuscript, the papers labelled with bold are included in our new version. Indeed there are many molecular data in prefer of our results, in case of not limiting the length of the manuscript, the authors would be happy to include a separate paragraph concerning molecular data gained in correlation with health and nutrition.

Dietary Implications of the Bidirectional Relationship between the Gut Microflora and Inflammatory Diseases with special emphasis on Irritable Bowel Disease: Current and Future Perspective. Nutrients 2023, 15, 2956. https://doi.org/10.3390/nu15132956

Zhou, Y., Sun, X., Yang, G., Ding, N., Pan, X., Zhong, A.,... Chai, X. (2023). Sex-specific differences in the association between steps per day and all-cause mortality among a cohort of adult patients from the United States with congestive heart failure. Heart & Lung, 62, 175-179. doi: https://doi.org/10.1016/j.hrtlng.2023.07.009

Zhang, Y., Zhao, C., Zhang, H., Chen, M., Meng, Y., Pan, Y., Zhuang, Q., & Zhao, M. (2023). Association between serum soluble α-klotho and bone mineral density (BMD) in middle-aged and older adults in the United States: a population-based cross-sectional study. Aging clinical and experimental research, 35(10), 2039–2049. https://doi.org/10.1007/s40520-023-02483-y

Bao, M., Luo, H., Chen, L., Tang, L., Ma, K., Xiang, J.,... Li, J. (2016). Impact of high fat diet on long non-coding RNAs and messenger RNAs expression in the aortas of ApoE(−/−) mice. Scientific Reports, 6(1), 34161. doi: 10.1038/srep34161

Chen, Y., Xiang, J., Wang, Z., Xiao, Y., Zhang, D., Chen, X.,... Zhang, Q. (2015). Associations of Bone Mineral Density with Lean Mass, Fat Mass, and Dietary Patterns in Postmenopausal Chinese Women: A 2-Year Prospective Study. PLOS ONE, 10(9), e137097. doi: 10.1371/journal.pone.0137097

The authors should also add some details about exercise as well.

Indeed physical activity is strongly correlated to health parameters. The following international study was added to Discussion:

A 2-year multilevel intervention, on energy balance-related behaviors among European families at risk for developing type 2 diabetes was conducted, based on self-reported physical activity, sedentary and eating behaviors in: Belgium, Finland, Greece, Spain, Hungary and Bulgaria [55]. Unfavorable intervention effects were found on the consumption of soft drinks and sugar-containing juices among Hungarian children and parents, while examining the intervention effects for the overall population and per country, 10 from the 112 investigated outcome variables were improved in the intervention group compared to the control group. Besides diet the activity level of elderly leads to healthier ageing. Daily step counts are associated with all-cause mortality in patients with congestive heart failure, 5581 daily steps were associated with a decreased risk of all-cause mortality in patients [56].

Please add some latest literature / references to your article.

Thank you very much for your suggestion! Many articles are included in the manuscript, hoping, we have chosen the adequate ones.

Figures quality should be improved.

Thank you very much for your suggestion! The figure was newly created. The authors hope that it is more informative and its quality is much better!

Conclusion should be added to this study.

Thank you very much, for your suggestion! The following conclusion was added to the manuscript:

Nutritional habits of Hungarian elderly, retired population, similarly to other data, are strongly correlated to the socioeconomic status, original educational level, lifestyle habits of persons [2,4, 29]. There are sex differences in the possibility of malnutrition (in favor of females), which correlates with the loss of appetite and meal consumption. Males are having higher values of BMI, with more cases of overweight or obesity[40]. The difference in food product consumption partly correlating with the fact that M are eating more often out of home, where meals contain much more vegetables, fruits than homemade food. Changes in the socioeconomic status of this population during the last decades are visible in the increases of the amounts consumed, changing from less protein and more vegetables in the diet to the opposite. The increase of CH consumption is measurable in the type of food consumed also, more white bread, potatoes, pasta. In the traditional Hungarian diet, like in many LMIC, legumes are very important sources of CH and proteins, further increasing the amount and energy consumed. Water consumption is generally very low in accordance with other data, one third of the studied population drinks less, than 1Liter, 15% of responders drink only half a Liter water daily, M are drinking significantly more. Although very high percentage of Hungarian elderly have two, or three chronic diseases, visiting medical centers, meeting professionals regularly, their general knowledge of healthy diet is low.

The language and style of the manuscript was improved. A native English speaker edited and reviewed our manuscript. We hope we were able to enhance clarity and readability of our work.

Finally, we greatly appreciate your detailed work and feedback on our manuscript! We hope that our revised manuscript will reach the standard of Nutrition Journal and will be satisfactory for publication!

Reviewer 2 Report

Comments and Suggestions for Authors

The study employs a multi-faceted approach, combining the Mini Nutritional Assessment questionnaire, on nutritional habits, and a 3-day food diary. This comprehensive methodology allows for a detailed analysis of nutritional patterns in the elderly population. The study goes beyond simple caloric intake, focusing on macronutrients, water consumption, and the frequency of fruit and vegetable consumption. This provides a nuanced understanding of dietary habits and potential nutritional factors influencing health in the elderly.

The study's sample size (179 participants) might be considered relatively small, and its representativeness for the entire Hungarian elderly population could be questioned. A larger, more diverse sample could enhance the generalizability of the findings.

The study relies on self-reported data from questionnaires and feeding diaries, which might be subject to recall bias and social desirability bias. Participants might underreport or overreport certain aspects of their dietary habits. What care has been taken for minimizing this effect.

While the study mentions the exploration of nutritional factors leading to NCDs, it doesn't provide specific details on the types of chronic diseases or associations with particular dietary patterns. A more detailed analysis of these relationships could enhance the study's impact.

The study does not appear to delve into socioeconomic factors, which can significantly influence dietary habits. A more comprehensive understanding of the relationship between nutrition and socioeconomic status could add depth to the findings.

In conclusion, while the study provides valuable insights into the nutritional habits of the Hungarian elderly population, addressing the highlighted limitations could enhance the robustness and applicability of the findings.

Comments on the Quality of English Language

There are many typographical errors throughout the manuscript, I assume  the typographical errors are due to translation to English.

For examples: page 3, second paragraph, "In this softver the USDA......", it is software.

Author Response

 Dear Reviewer,

Thank you very much for your work your very helpful feedback a constructive suggestions!

Indeed with your feedback and suggestions we have the opportunity to improve our manuscript Below are the actions, comments/thoughts we have taken to improve the quality of the manuscript:

The study employs a multi-faceted approach, combining the Mini Nutritional Assessment questionnaire, on nutritional habits, and a 3-day food diary. This comprehensive methodology allows for a detailed analysis of nutritional patterns in the elderly population. The study goes beyond simple caloric intake, focusing on macronutrients, water consumption, and the frequency of fruit and vegetable consumption. This provides a nuanced understanding of dietary habits and potential nutritional factors influencing health in the elderly.

The study's sample size (179 participants) might be considered relatively small, and its representativeness for the entire Hungarian elderly population could be questioned. A larger, more diverse sample could enhance the generalizability of the findings.

Thank you very much for your suggestion! Indeed, considering large scale studies, our data are not representative, however one must consider that in Hungary volunteers are really sporadic in case of any Health or Fitness studies, especially among elderly. This population is living with many chronic diseases and not willing to cooperate in such studies. Unfortunately paying to subjects is forbidden by law. So in our case the number of subjects is relatively big, it is filling many gaps in Hungarian terms.

The study relies on self-reported data from questionnaires and feeding diaries, which might be subject to recall bias and social desirability bias. Participants might underreport or overreport certain aspects of their dietary habits. What care has been taken for minimizing this effect.

Thank you very much for your comment! Indeed self reported data are often not reliable, for the above mentioned reasons. For this reason, we stricktly used the exclusion criteria (see below and the text) and we worked with trained helpers in each case discussing detailes with all the responders, always the same way and describing the importance of appropriate data.

Exclusion criteria were: dementia, bad health status/sickness with special dietary needs, dialysis, or special feeding through the nose. Those who had major sensory disabilities (poor hearing, or vision) were also excluded from the study.

Data collection happened with the intention of collecting higher number of responders with useful and trustful results, so helpers were trained (nurses and physiotherapists, N=10) to ask and code the answers the same way, using the snowball method.

While the study mentions the exploration of nutritional factors leading to NCDs, it doesn't provide specific details on the types of chronic diseases or associations with particular dietary patterns. A more detailed analysis of these relationships could enhance the study's impact.

The reviewer is perfectly right in this fact, the suggestion is very important! Since there are many diseases mentioned in the introduction and the space is limited in a manuscript, we have included the following sentences in Introduction:

Osteopenia and osteoporosis are also important consequences of aging, and so are unfavorable lifestyle habits. In a Chinese postmenopausal population lean mass was the best determinant of bone mineral density [7]. Six dietary patterns were also identified but only cereal grains-fruits and milk-root vegetables patterns were associated with bone density of the spine and hip.

In case of possibility (room for a paragraph) the authors would be happy to introduce specific details of NCD in Hungary. Unfortunately there are many chronic diseases in the adult population, and especially in elderly, connected to unhealthy lifestyle habits.

The study does not appear to delve into socioeconomic factors, which can significantly influence dietary habits. A more comprehensive understanding of the relationship between nutrition and socioeconomic status could add depth to the findings.

Thank you very much for your suggestion! The following facts are included in the manuscript:

Higher sugar intake will increase calorie intake, the probability of overweight, obesity and development of noncommunicable diseases [1], just like socioeconomic status being one of the strongest predictors of obesity and unhealthy food environments [2]. Free sugar consumption and NCD are higher in the low- or middle-income countries (LMIC), like Hungary

There are many data discussing socioeconomic determinants in dietary patterns [4], in LMIC there was a strong correlation between fruit and vegetable, just like healthy food consumption and income of the population.

Lower socioeconomic status subjects intend to eat larger portions of unhealthy snacks, rather than small ones, leading to a potential 15-22% increase in energy intake (Best et al, 2019). In a Hungarian study socioeconomic differences in nutritional status was analyzed and "unhealthy diet" was 70.6% among Hungarians [29].

 Emotional eating was found to be rather caused by experiencing financial strain, rather than by traditional socioeconomic status dimensions in women, while restrained eating was associated with higher household income level in women and with higher occupational position in men [33]. In Hungary elderly living from low income face challenges, including managing diet. Unhealthy, energy-dense foods are purchased more frequently in these populations [34].

In conclusion, while the study provides valuable insights into the nutritional habits of the Hungarian elderly population, addressing the highlighted limitations could enhance the robustness and applicability of the findings.

Thank you very much for your comment! To introduce the importance of data collected in this study the followings were included in the Discussion section of the manuscript:

 A 2-year multilevel intervention, on energy balance-related behaviors among European families at risk for developing type 2 diabetes was conducted, based on self-reported physical activity, sedentary and eating behaviors in: Belgium, Finland, Greece, Spain, Hungary and Bulgaria [55]. Unfavorable intervention effects were found on the consumption of soft drinks and sugar-containing juices among Hungarian children and parents, while examining the intervention effects for the overall population and per country, 10 from the 112 investigated outcome variables were improved in the intervention group compared to the control group. Besides diet the activity level of elderly leads to healthier ageing. Daily step counts are associated with all-cause mortality in patients with congestive heart failure, 5581 daily steps were associated with a decreased risk of all-cause mortality in patients [56].

We also included limitations and a Conclusion section at the end of the manuscript.

The limitations of our study are partly due to the fact that low number of volunteers are responding in a study. One cannot reach the general adult Hungarian population in high number, especially among elderly. Although the educational level might strongly modify obtained data, in Hungary most of the population will learn about healthy food and diet. Since socioeconomic status limits the quality of food consumed, indeed, over reporting might shade the results obtained.

Conclusion

Nutritional habits of Hungarian elderly, retired population, similarly to other data, are strongly correlated to the socioeconomic status, original educational level, lifestyle habits of persons [2,4, 29]. There are sex differences in the possibility of malnutrition (in favor of females), which correlates with the loss of appetite and meal consumption. Males are having higher values of BMI, with more cases of overweight or obesity[40]. The difference in food product consumption partly correlating with the fact that M are eating more often out of home, where meals contain much more vegetables, fruits than homemade food. Changes in the socioeconomic status of this population during the last decades are visible in the increases of the amounts consumed, changing from less protein and more vegetables in the diet to the opposite. The increase of CH consumption is measurable in the type of food consumed also, more white bread, potatoes, pasta. In the traditional Hungarian diet, like in many LMIC, legumes are very important sources of CH and proteins, further increasing the amount and energy consumed. Water consumption is generally very low in accordance with other data, one third of the studied population drinks less, than 1Liter, 15% of responders drink only half a Liter water daily, M are drinking significantly more. Although very high percentage of Hungarian elderly have two, or three chronic diseases, visiting medical centers, meeting professionals regularly, their general knowledge of healthy diet is low.

Comments on the Quality of English Language

There are many typographical errors throughout the manuscript, I assume  the typographical errors are due to translation to English.

Thank you for your suggestion! The language and style of the manuscript was improved. A native English speaker edited and reviewed our manuscript. We hope we were able to enhance clarity and readability of our work.

For examples: page 3, second paragraph, "In this softver the USDA......", it is software.

Thank you for your kind help! We hope that typing errors were deleted from the text.

Finally, we greatly appreciate your detailed work and feedback on our manuscript! We hope that our revised manuscript will reach the standard of Nutrition Journal and will be satisfactory for publication!

Round 2

Reviewer 1 Report

Comments and Suggestions for Authors

The authors have significantly revised the manuscript and can be accepted for publication. 

Author Response

Thank you.

Reviewer 2 Report

Comments and Suggestions for Authors

Authors responses are satisfactory. All changes made to this version of manuscript can be accepted for publication.

Author Response

thank you.